# Advancements in Research and Applications of PP-Based Materials Utilizing Melt-Blown Nonwoven Technology

**DOI:** 10.3390/polym17081013

**Published:** 2025-04-09

**Authors:** Ziyang Fang, Jie Wang, Sijia Xie, Zhouyang Lian, Zhengwei Luo, Yan Du, Xueying Zhang

**Affiliations:** School of Environmental Science and Engineering, Nanjing Tech University, Nanjing 211816, China; fzy2@njtech.edu.cn (Z.F.); 202361202046@njtech.edu.cn (J.W.); 202261202036@njtech.edu.cn (S.X.); luozw2015@njtech.edu.cn (Z.L.); duyan@njtech.edu.cn (Y.D.); xueyingzhang@njtech.edu.cn (X.Z.)

**Keywords:** melt-blown nonwoven material, polypropylene, melt-blown nonwoven process technology, media filtration, adsorption separation, battery separator

## Abstract

Melt-blown nonwoven materials have demonstrated significant advancements in a multitude of industrial sectors, mainly due to their high production efficiency, extensive specific surface area, and narrow aperture. The demand for melt-blown nonwoven materials has increased further in recent time, particularly in the wake of the novel coronavirus (COVID-19) pandemic. Polypropylene (PP) is extensively used in production and research due to its low cost, low weight, and easy processing, and the melt-blown materials made from it share similar characteristics. We systematically summarize the research advancements of melt-blown nonwoven technology and applications of PP-based melt-blown materials over the last few years. First, the principles and processes of melt-blown nonwoven that govern the production of micro/nano fibers are described. Then the raw materials and process technology of melt-blown are reviewed. After these, we highlight the use of PP-based melt-blown materials in key fields, including media filtration, oil–water separation, heavy metal ions removal, organic pollutants removal and battery separator. Finally, we summary and suggest some potential future research directions of melt-blown nonwoven technology and PP-based melt-blown materials.

## 1. Introduction

The history of the melt-blown nonwoven process can be traced back to 1954, when the U.S. Naval Institute developed the melt-blown nonwovens technology with the objective of collecting radioactive particles in the upper atmosphere. At that time, the process could produce microfibers or fibers with diameters of less than 10 μm. In the 1960s, Esso Research and Engineering (subsequently renamed ExxonMobil) enhanced and expanded its production lines and mold widths, thereby achieving the successful production of polypropylene (PP) microfibers at a low cost. Subsequently, research on melt-blown nonwoven processes and equipment has progressed at a rapid pace, and the application areas of melt-blown technology have been expanding [1]. As illustrated in Figure 1, the number of research papers in this field has been increasing annually for the past two decades, reaching a peak of over 10 papers per year in 2023. This growth can be attributed to the emergence of the New Crown epidemic in 2020, which led to a surge in research activity in this area.

Despite the rapid advancement of melt-blown nonwoven technology, there has been a paucity of reviews on this topic in recent years. This paper, however, employs CiteSpace software (v.6.3.R1 (64-bit)) to synthesize the findings from the core database of the Web of Science over the past five years and conduct keyword clustering and time mapping analysis [2]. For further details, please refer to Figure 2. The modular value Q = 0.778 indicates that this co-occurring network clustering structure has a high degree of significance. The average contour value S = 0.9177 indicates that the co-occurrence network clustering results have a high degree of confidence. The 298 keyword nodes are divided into 11 clusters, with the size of the keyword circle logo positively correlated with frequency. Each clustering module is labeled with a number from 0 to 11. The clustering structure depicted represents the research hotspots to a certain extent, which include: the following topics were identified: air filtration, melt blowing, polyolefins, composites, surface modification; mechanical properties, oil/water separation, antibacterial, Sound absorption and Selective. By further summarizing the clustering labels and their internal keywords, the research topics can be classified into three categories: raw materials for melt-blown nonwoven fabrics, melt-blown nonwoven fabrics technology, and applications of melt-blown nonwoven fabrics.

Based on three research topics, this paper introduces the melt-blown process and the principle of micro/nano-fiber manufacturing, reviews the development of melt-blown raw materials and melt-blown process technology. The paper also considers the applications of PP-based melt-blown nonwoven materials, including filtration, oil–water separation, heavy metal ion adsorption, organic pollutant removal, healthcare and energy field. It analyses the current problems and development directions in these areas, to provide a certain reference for the research and application of melt-blown nonwoven materials.

## 2. Melt-Blown Spinning Principles and Processes Description

Melt-blown nonwoven technology, as the core process for direct polymer web formation, relies on high-speed hot air to stretch molten polymer fluid into ultrafine fiber networks. As shown in Figure 3, the melt-blown spinning system primarily comprises a screw extruder, spinning assembly, and filament stretching/collection device. Preprocessed polymer raw materials (such as PP) are fed into the screw extruder, where they undergo melting and plasticization through zoned heating and mechanical shear. A gear metering pump then precisely controls the flow rate to the spinning assembly [3,4,5]. The spinneret—a critical component for fiber uniformity—typically features orifices of 0.25–0.4 mm in diameter with a length-to-diameter ratio >10. High-temperature, high-velocity air jets $\left(60{}^{\circ}\right.$ angular configuration on both die sides) induce turbulent flow fields for secondary stretching of molten streams, refining fiber diameters to 1–5 μm. During web formation, the die to collector distance (DCD) regulates fiber cooling time and self-bonding degree, determining web loftiness and mechanical strength, while the airflow velocity of the vacuum suction device governs fiber deposition density [6,7,8,9].

Key process parameters can be categorized into material properties, process variables, and equipment specifications. For raw materials, polymer melt flow index is pivotal; higher index enhances melt fluidity but reduces fiber strength due to lower molecular weight, necessitating narrow molecular weight distribution materials to balance spinnability and mechanical performance [10]. Process-wise, synergistic effects of hot air velocity and temperature are critical. Yesil, Y. et al. [11] systematically investigated the impacts of die temperature, air pressure, and DCD on fiber diameter. At elevated die temperatures, reduced melt viscosity improves fluidity and post-die attenuation, yielding finer fibers. Fiber diameter decreases with increasing air pressure and DCD until reaching plateaus at 35 kPa and 35 cm, respectively, beyond which, sensitivity diminishes.

Common methods for manufacturing PP micro/nano-fiber include flash evaporation [12] and spunbonding [13]. Flash-spun PP fabric boasts high strength and light weight, yet their production necessitates the use of a substantial number of toxic solvents. Spunbonding, while capable of providing high strength to melt-blown raw materials, results in fabrics with poor barrier properties. Traditional methods also encompass dry spinning and wet spinning. In dry spinning, melt-blown raw materials is combined with a highly volatile solvent and fed into a twin-screw extruder. Most of the solvent evaporates during extrusion under a hot nitrogen purge, leaving dried fine fibers that are then stretched and wound to form the final fibers. However, fibers produced by this method often retain a small amount of solvent residue, which is challenging to remove. The wet spinning process initially mixes PP with olefins to create a homogeneous solution at high temperatures, forms a gel film post-extrusion and cooling, and subsequently subjects the gel film to extraction and stretching treatments. This process is relatively complex. Additionally, electrospinning technology presents an effective method for preparing PP micro/nano-fiber, although the process is inefficient and typically requires the use of a large quantity of toxic solvents [14]. In comparison, although the production of nanofibers through the melt-blowing process requires process optimization, it still boasts the advantages of high efficiency and low cost compared to electrospinning, while also avoiding the generation of a large amount of toxic vapor waste [15,16].

Currently, PP nonwoven materials produced through melt-blown technology boast advantages such as fine fiber diameter, large specific surface area, high porosity, low cost, and environmental friendliness. They have attracted widespread attention for their applications including filtration and adsorption. As a result, the melt-blown process has become one of the leading methods for the preparation of micro/nano-fiber materials [17]. Recent advancements, including bicomponent melt-blowing and electret melt-blown technologies, further expand functional boundaries. However, challenges persist, such as strength limitations from low fiber orientation, trade-offs between high melt flow rates and energy consumption, and process instability. Future efforts may focus on multi-physics coupling modeling and online monitoring for dynamic parameter optimization, advancing high-end applications in organic pollutant removal, medical protection, and energy filtration systems [18,19,20,21].

## 3. Research Progress of Polypropylene-Based Melt-Blown Nonwoven Materials

### 3.1. Raw Materials

Given the numerous advantages of PP and its melt-blown materials, most commercial development of melt-blown products and processes, as well as academic research, has concentrated on PP polymers [22,23,24]. Pure PP can be processed directly into melt-blown nonwovens, but the formed fabric usually undergoes modification to enhance material properties before application. Srikhao, N. et al. [23] had developed a new filter by spraying green-synthetic nano silver onto formed melt-blown PP fabric. Its filtration efficiency of various particle sizes is increased by 27.95–447.04% compared with the pure PP, and the filtration efficiency of 0.3 μm particles (including PM1.0 and PM2.5) exceeds 97%. Eticha, A. et al. [24] successfully manufactured biodegradable and biocompatible filters from PP and polylactic acid polymer using the melt-blowing process. The soil embedding experiment showed that the prepared filter materials were degraded to varying degrees within 90 days. Qin, Y. et al. [25] used plasma grafting polymerization technology combined with ring-opening amination to modify PP melt-blown fiber as a boron adsorbent. At pH value of 6, the maximum saturation adsorption capacity of boron is 18.03 ± 1 mg/g, and it shows excellent selectivity in the presence of multiple cations. Lam, M. et al. [26] grafted sodium polystyrene sulfonate onto the formed large entangled fiber PP membranes in one step by ultraviolet radiation, and the resulting material exhibited chemical biological reactivity, which can promote cell adhesion, and the discovery is conducive to the future application of melt-blown materials in the biological field. On the other hand, the raw materials of PP-based melt-blown nonwovens will not be limited to pure PP. Xie, S. et al. [27] prepared modified PP by optimized grafting process and then prepared functional fibers by melt-jet spinning to adsorb indigo in wastewater, whose adsorption capacity could remain 91.22 mg/g after eight regeneration and reuse cycles.

In addition to modifying PP raw materials, it is also possible to directly incorporate other polymers for blending melt-blowing processes. Peng, M. et al. [28] blended thermoplastic polyurethane (TPU) and PP to produce composite nonwoven materials: the resulting materials were soft and elastic with a certain tensile resistance. SEM observation showed that the single fiber in the nonwovens formed a sea-island structure in which TPU served as the ’island’ phase and PP served as the ’sea’ phase. This structure was identified as the reason for the good elasticity of the composite nonwoven materials. Eticha, A.K. et al. [29] developed a flexible and high-filtration performance PP-TPU textile material by melt-blowing. PP’s tensile strength increased by 72.22%, and elongation improved by 38.1% with the addition of 20 wt.% TPU.

Specific functional modification additives can also be used as added raw materials. Liu, N. et al. [22] synthesized the organic-inorganic synergistic UV-stabilized material via the co-precipitation method. This material was subsequently incorporated into PP matrix through blending melt-blown process, resulting in the development of a novel composite fiber with UV aging resistance. The thermal decomposition temperature of this composite fiber was found to be 38 °C higher than that of pure PP fiber, and the fiber’s strength retention rate remained as high as 94.56% after 6 days of aging. Ma, T. et al. [30] developed a hybrid microsphere structure material by blending nano ZnO masterbatch with PP particles, which exhibits superior mechanical properties, crystallinity, and thermal stability compared to pure PP. Besides, the material exhibited the outstanding performance and durability (the breaking strength of 2.43 N, the air permeability of 537mm/s, the filtration efficiency of 97.25%, and the filtration resistance of 28.38 Pa). Shen Y. et al. [31] fabricated a polymer composite material consisting of dispersed rice husk carbon and PP via melt blending process. This approach not only enhanced the rigidity and dimensional stability of the material but also increased the initial decomposition temperature of the PP-based material by 130 °C.

### 3.2. Development of Melt-Blown Manufacturing Technology

#### 3.2.1. Bicomponent Melt-Blown Technology

To address the issues of low strength and coarse fiber diameter in single-component melt-blown products, bicomponent melt-blown technology has been developed. This technology involves melting and extruding two or more polymers, blending or arranging them in a side-by-side configuration through specially designed spinneret holes, and then stretching them into ultra-fine fibers using high-speed hot air to form a nonwoven fabric with a specific microstructure. Bicomponent melt-blown technology can achieve different functionalities through the combination of multiple polymers with complementary properties or the addition of functional additives. A typical example is the incorporation of PP to reduce costs and improve chemical stability and processing performance. However, it is often necessary to add compatibilizers or optimize processing conditions to control interfacial compatibility between different components and prevent fiber breakage. Currently, the development of high-functionality combinations is still in the exploratory stage, and industrialization remains challenging [32,33,34].

#### 3.2.2. Melt-Blown Electret Technology

Melt-blown electret technology can overcome the limitations [35] of traditional melt-blown nonwoven filtration materials in capturing airborne hazardous substances. Melt-blown electret technology employs an electret charging process infuse melt-blown nonwoven fabrics with persistent electrical charges. This technology finds widespread application in N95 respirator materials and high-efficiency particulate air filtration systems [36]. Here, the electret is a dielectric that maintains long-term charge. Electric electret (EE) and water electret (WE) are currently widely used.

For EE, during the polymer melt extrusion phase, high-voltage corona discharge creates deep charge traps on the fiber surface, which retain electrical charges after fiber solidification, forming a stable electrostatic field. When airflow containing particles passes through, electrostatic attraction synergizes with mechanical interception to significantly enhance filtration efficiency and enable the melt-blown nonwoven material with exceptional capability to capture fine particulate matter. EE melt-blown nonwoven fabrics offer advantages such as low production costs and ease of manufacturing. However, elevated ambient temperatures and humidity levels can induce charge decay, and the EE process generates ozone, which are challenges that need to be addressed in industrial settings [37,38,39,40].

Figure 4 [40] depicts one of EE methods, the bielectret corona charging treatment of melt-blown nonwoven fabric, which differs from the process illustrated in Figure 3. When a high positive voltage is applied to the electrode at the end, the electric field causes the neutral molecules in the air to be ionized into $H^{+},{\mathrm{NO}}^{+}$, and ${\mathrm{NO}}_{2}^{+}$ ions. These positive ions move towards the grounded electrode, forming a surface charge on the nonwoven fabric. Accordingly, the dual standing electrode technique can compensate for the deficiencies of the single standing electrode method, thereby facilitating a more stable and efficient standing electret effect.

For WE, the electret is activated through the infiltration and friction of pure water on the melt-blown nonwoven fabric. During this process, charges are rearranged and uniformly distributed. Subsequently, after passing through a hot air device, part of the surface charges on the WE melt-blown nonwoven fabric are removed, leaving primarily deep trap charges and dipole charges. Compared with EE melt-blown nonwoven fabrics, WE fabrics exhibit relatively higher filtration efficiency and lower air resistance. Additionally, the use of pure water in the WE process enhances its environmental friendliness. However, potential drawbacks include high water consumption and increased production costs [36,39,41].

#### 3.2.3. Intercalated Melt-Blown Technology

The melt-blown fabric, self-thermally bonded by ultrafine fibers, features a tight structure and high filtration efficiency, but this also leads to increased filtration resistance and correspondingly reduced dust-holding capacity. To address these issues, 3M in the United States has developed the Intercalated melt-blown technology [42].

This technology incorporates an additional carding machine that combs short fibers into a web. During the process where the polymer melt is drawn by high-speed hot air streams from both sides of the melt-blown die towards the winding net bag, highly crimped short fibers are uniformly interspersed in the polymer melt jet in the form of individual fibers by crosswinds provided by a blowing device. Leveraging the advantages of short fibers’ good elasticity and high rigidity, this adjusts the pore structure of the melt-blown fiber membrane, improves its elasticity and pressure resistance, and enhances the interception effect, significantly boosting the interlayer melt-blown fabric’s barrier performance against particles. Data models indicate that in the interlayer melt-blown process, there is a strong positive correlation between the receiving distance and both thickness and porosity, while a negative correlation exists between the hot air velocity and the compression resilience rate. Intercalated melt-blown technology can optimize its process through algorithmic control, demonstrating high flexibility in its application. However, correspondingly, it requires high precision equipment and extensive operational experience, and some performance indicators are subject to limitations [42,43,44].

#### 3.2.4. Nanoval Split Spinning Technology

Nanoval split spinning is a technology for spinning submicron fibers based on melt-blown, as proposed by the German Nanoval company, and its process flow diagram is shown in Figure 5. As a relatively new and complex processing technology, it sits between the melt-blown and spunbond processes, combining elements of the “Biax fiberfilm die” and “metal injection molding technology”. It enables the melt to exit the nozzle along with air, utilizing residual heat to bond into a fabric. This technology is suitable for manufacturing nonwoven fabrics from a variety of complex polymers with viscosity ranging from 2 to 18 Pa*s and melt flow indices between 35 g/10 min and 1200 g/10 min, offering high selectivity and flexibility. Furthermore, for PP, fibers with diameters ranging from below 1 μm to 40 μm can be obtained without altering the spinning equipment. The resulting nonwoven fabric exhibits properties of both melt-blown and spunbond fabrics, making it ideal to produce high-performance nonwovens in small batches and with diverse varieties. Compared to melt-blown production lines under the same processing conditions, it can reduce energy consumption by approximately 60%. Regrettably, the initial investment cost for this process is relatively high, and further optimization is needed to address the sensitivity to process parameters and the constraints of high-temperature processing. Additionally, the issue of fiber deposition during the process may affect the uniformity of the nonwoven fabric. Overall, the Nanoval split spinning technology is still in the research stage [45,46].

In recent years, the melt-blown process has been continuously innovated and optimized, and there have been new processes and technologies in high pore density melt-blow spinneret, nano melt-blow fiber process and technology, twin-shaft melt-blow system, melt-blow spun bonding and electro-centrifugal spinning [47]. Each process of interpenetration, to the direction of hybrid, composite development is the current trend of the development of nonwovens, especially the composite between the processes is increasingly gaining attention.

### 3.3. Applications of Polypropylene-Based Melt-Blown Nonwoven Material

The selection of melt-blown raw material and the melt-blowing process is geared towards practical functional applications. The good material properties and processing performance of PP facilitates its large-scale production. The typical microporous structure of nonwoven materials and the small pore size and large specific surface area resulting from fine fibers enable PP-based melt-blown materials to possess good physical properties while serving as an excellent matrix for functional modification, with research ongoing in multiple application fields.

#### 3.3.1. Filter Material

The properties of PP-based nonwoven materials make them well-suited for various filtration applications, including fuel purification, air filtration, solid-liquid separation and so on [24,48].

For instance, Zhao, Y. et al. [49] developed a bicomponent melt-blown fuel filtration material by combining a PBT/PP melt-blown layer with a cellulose nanocrystal-doped cellulose wood fiber layer. This composite achieved a filtration efficiency of 99.90% for particles larger than 14 μm and 99.52% for those exceeding 4 μm, with a dust-holding capacity of 27.63 mg/cm^2^, indicating its suitability for long-term fuel filtration. Wang, X. et al. [50] developed a three-layer composite filter using PP/PE bicomponent melt-blown nonwovens. The material exhibited an exceptional pure water flux of 18,447.86 L/(m^2^·h) and a 99.99% separation efficiency for slag powder, with a filtrate turbidity of 7.9 NTU. Additionally, it showed strong regeneration capability and mechanical stability, making it ideal for industrial-scale applications. In the study conducted by Zhang, J. et al. [51], commercial PP melt-blown nonwoven was oxidized by acid potassium permanganate solution while nano- ${\mathrm{MnO}}_{x}$ particles were deposited on the fiber surface. The as-prepared membranes exhibit superhydrophilicity and underwater superoleophobicity with high separation efficiencies of over 99% for complex oil/water mixtures, and the permeation flux could reach 61,177 L/(m^2^·h). Li M. et al. [52] had produced a reusable composite filter material utilizing polytetrafluoroethylene microporous film and PP-based melt-blown nonwoven fabric. The bicomponent melt-blown nonwoven filter material exhibits strong bond strength and a stable structure, with a filtration efficiency of up to 99.95% and a filtration resistance of 350 Pa. Zhao, L. et al. [53] further explored nonwoven filtration materials in the coffee industry, fabricating a double-layer composite by integrating PP melt-blown nonwovens with V/ES fiber layers. The study revealed that microfiber density influenced filtration speed, with denser layers slowing the process. The material also exhibited improved hydrophilicity after filtration, highlighting its potential for specialized applications.

Nanofibers exhibit smaller resistance, higher permeability, and greater porosity than microfibers in filtration applications, resulting in superior filtration efficiency under the same pressure drop [54]. Sun, F. et al. [55] designed a multi-scale membrane composed of polyurethane, polyvinylidene fluoride, and PP. The electret membrane exhibited a filtration efficiency of 96.72% and a quality factor of $0.04167\;{\mathrm{Pa}}^{-1}$, demonstrating the advantages of combining microfibers and nanofibers to improve filtration performance and reduce pressure drop. Zakaria, M. et al. [56] employed PP nanofibers with an average diameter of 228 nm for bicomponent blending and melt-blowing, and then prepared nanofiber films by wet layer method. The resulting membrane exhibited a porosity of 70.8%, an average pore size of 0.087 μm with narrow size distributions.

#### 3.3.2. Oil Absorbent Material

PP fiber film has good physical properties and a certain polarity, which is suitable for oil absorption material. With the increase of application demand, researches have modified PP-based melt-blown materials to obtain better oil absorption performance [57,58].

Sokolovic, S.S. et al. [59] studied PP fibers as a bed layer coalescent and found that for PP materials, a large number of capillaries are formed at a higher bed porosity, which is conducive to the effective coalescence of oil droplets and thus better separation of oil droplets. On this basis, waste PP bags used to package vegetables can be transformed into rectangular cross-section fibers as a filter medium, which can reduce the oil concentration in the effluent to less than 15mg/L. Moreover, these raw materials without any cost reduce environmental pollution. Luo, Q. et al. [60] synthesized organically modified saponite via solution polymerization and blended it with PP for by bicomponent melt-blown technology. The resulting nanocomposite fiber film exhibits excellent oil adsorption capacity, thermal stability, and reusability, whose adsorption for xylene and kerosene reached a maximum of 15.84 and 22.84 g/g, respectively. In the treatment of coal tar wastewater, the removal efficiency of oil can reach 62.9%, and it could still maintain a removal rate of 51.17% after five experimental cycles. After hydrophilic modification of commercial PP melt-blown nonwovens, Tang, Z. et al. [61] incorporated layered Co-Al metal hydroxides in situ. The resulting composite exhibited an adsorption efficiency exceeding 99.00% for 50 mg/L Congo red within 30 min. Additionally, the separation efficiency for n-hexane/water mixtures reached 99.33%, with the efficiency remaining above 97.41% during 80 repeated cycles. Sun, Y. et al. [62] conducted a hydrophilic modification of PP melt-blown nonwoven materials and subsequently synthesized cobalt hydroxyl carbonate on the surface of the modified materials. When n-hexane is employed as the oil phase, the underwater oil contact angle of the resulting material can reach over 140.63°, with an oil–water separation efficiency exceeding 98.76%. Furthermore, after 50 cycles of oil–water separation, the efficiency remains no less than 97.02%.

#### 3.3.3. Heavy Metal Ions Adsorption

Melt-blown PP fiber has some inherent limitations, including low polarity, inadequate hydrophilicity, especially the absence of active functional groups, present challenges in its direct application for the removal of heavy metal ions from wastewater. Researchers have conducted investigations to address these shortcomings. Chen, H. et al. [63] used lab made fibers to prepare PP-g-GMA-DETA fibers based on Pearson’s HSAB theory to remove ${\mathrm{Cd}}^{2+}$ and ${\mathrm{Pb}}^{2+}$ from wastewater. Its maximum adsorption capacities of ${\mathrm{Cd}}^{2+}$ and ${\mathrm{Pb}}^{2+}$ in wastewater reached 41.87mg/g and 31.40 mg/g, respectively. Liu, C. et al. [64] prepared a PP hollow fiber membrane with selective adsorption properties. The prepared fiber exhibits high adsorption of Hg (II) ions in aqueous solution, with a maximum theoretical adsorption capacity of 0.854 mmol/g. In order to remove ${\mathrm{Cd}}^{2+}$ and ${\mathrm{Cu}}^{2+}$ from water, Chen, R. et al. [65] utilized ultrasonic waves to promote chemical grafting. Benzoyl peroxide was employed as the initiator to graft methacrylic acid onto the surface of formed melt-blown PP fibers in an aqueous suspension system. Weakly acidic cation exchange PP-g-MAA fibers with a grafting rate of 7.5% were prepared. The adsorption and removal rates of ${\mathrm{Cd}}^{2+}$ and ${\mathrm{Cu}}^{2+}$ were 90% and 95%, respectively. Guo. M. et al. [66] synthesized PP chelate fibers grafted with acrylic acid and acrylamide side chains by traditional melt-blown process and utilized them as adsorbents for the selective removal of Pb (II) ions from aqueous solutions. The time to reach half of the adsorption capacity of the fiber was 6.2 min. However, the process exhibited susceptibility to interference from ${\mathrm{Ca}}^{2+}$ ions. Luo, Z. et al. [67] employed the technique of suspension graft copolymerization of water and solid to modify granular PP with dual monomer glycidyl methacrylate and acrylamide. Subsequently, they produced Cr(VI)-imprinted fibers through multi-component melt-blown process. The preparation principle is illustrated in Figure 6. The prepared Cr(VI)-imprinted fibers exhibit excellent selectivity for Cr(VI), with a maximum adsorption capacity of 43.2mg/g at pH=3.

#### 3.3.4. Organic Pollutant Removal

In the context of global industrialization, the emission and accumulation of organic pollutants have emerged as a significant concern on a global scale. The distinctive benefits of adsorption technology in the domain of organic pollutant removal have positioned it as a prominent area of research. The advancement of melt-blown nonwoven technology has not only facilitated the development of highly effective adsorption materials but also served as excellent supports for organic catalysts [68,69].

Luo, Z. et al. [68] produced reactive macromolecular radicals on the surface of homemade melt-blown PP fibers utilizing argon plasma irradiation, subsequently preparing surface-modified PP fibers PP-g-St through the in situ grafting of styrene monomers. At a grafting rate of 5.7%, the adsorption capacities of benzene, toluene, and xylene reached 18.6,16.8, and 13.4g/g, respectively. Werner, *Ł*. et al. [69] treated PP melt-blown nonwoven materials made by traditional melt-blown process with isopropanol to improve its hydrophilicity, then using hydrothermal chemical bath deposition method to cover the fiber surface with ZnO nanorods. It is possible to control the structures of obtained ZnO via the concentration of reagents used in the synthesis. The photocatalytic removal rate of methylene blue by the resulting ZnO/PP material is more than 60% after 6 h. Lian, Z. et al. [70] employed PP as the matrix, AA and MAH as the functional monomers, and prepared PP-g-(AA-MAH) fibers via the suspension grafting and traditional melt-blown processes. This resulted in a significant enhancement in the hydrophilicity of the fibers, with an aniline adsorption capacity reaching 42.2mg/g. Li, B. et al. [71] introduced carboxyl group on the surface of commercial PP fiber by radiation grafting polyacrylic acid, and then reacted it with ${\mathrm{Fe}}^{3+}$ ions to form a complex Fe-PAA-g-PP on the surface of the fiber. Under ultraviolet radiation, the degradation rate of dye reactive red 195 was close to 90%. Liu, Y. et al. [72] employed terephthalic acid as an organic ligand and ferric trichloride hexahydrate as the central metal source, and combined the solvothermal method with the impregnation process to synthesize ${\mathrm{TiO}}_{2}/\mathrm{MIL}$-88B(Fe)@PP composite melt-blown non-woven materials in situ on the surface of homemade PP melt-blown materials. Upon exposure to visible light, the degradation rates of methyl blue, acid orange 7, and acid red 73 all exceeded 80%. The degradation rate of methyl blue reached 86%, while that of Rhodamine B was 59%.

In addition, several studies also have explored the immobilization of organic catalysts onto the surface of fiber fabrics. Wang, W. et al. [73] employed a one-step in situ growth method under ultrasonic conditions to load photocatalysts rapidly onto homemade PP melt-blown nonwoven fibers, which solved the problem of nanocatalysts agglomeration. Upon exposure to visible light, the degradation of Congo red could reach 99.1% within 100 min and 99.6% within 120 min, meanwhile, the photodegradation still maintained about 85% after 5 cycles. Zhang, D. et al. [74] chemically adsorbed ${\mathrm{MnO}}_{2}$ onto the surface of modified PP melt-blown non-woven fabric. Under the conditions of 1 mg/mL${\mathrm{MnO}}_{2}$ loading concentration, pH = 3 and 25 °C, the removal efficiencies of Rhodamine B, methyl blue, methyl orange, and carmine in water all reached 99.00%. Notably, even after twenty adsorption–desorption cycles, the removal efficiency of Rhodamine B remained consistently around 99.00%.

#### 3.3.5. Medical and Health Material

The EU standard EN 14683 [75] distinguishes between fabric masks, which are made of textiles, and medical surgical masks, which are made of multi-layer nonwoven materials. For the mask to be suitable for medical purposes, it must meet two basic requirements. First, the materials must provide an effective barrier against the spread of droplets (≥5 μm). Secondly, the components of the mask should not interfere with normal breathing [76,77]. Čepič, G. et al. [78] investigated the effect of combining the spunbond and melt-blown processes on the functional properties of medical nonwovens. The find indicated that the five-layer composite comprising three layers of spunbond and two layers of melt-blown produced from PP fibers has the optimum properties for use as medical textiles, as it has the capacity to filter out pollutants while maintaining good air permeability.

Generally, PP nonwovens as medical materials are mainly concerned with the filtration efficiency of particulate matter, rather than pathogen control. Therefore, the surface of the fiber may harbor pathogenic bacteria, resulting in the risk of secondary infection. Through the modification of PP melt-blown cloth, it has antibacterial property, which can further meet the medical and health requirements of PP melt-blown cloth [79]. Liu, C. et al. [80] found melt-blown nonwovens, which have already been processed and formed, are more suitable for antibacterial modification via the spraying method. The electret melt-blown fabric prepared by the bicomponent melt-blown technique killed 99.99% of *E. coli* and *S. aureus* within 10 min contact time, while still maintained a filtration efficiency of 94.77% and a pressure drop of 76 Pa.

In the study conducted by Ma, Y. et al. [81], the muti-component melt-blown process was employed to graft methyl acrylamide as an acyclic halogen precursor onto PP backbone. These materials were then immersed in an active chlorine solution, resulting in the production of a halogen-bearing amine-type antimicrobial PP melt-blown fabric. The experimental results show that the fabric’s germicidal effect of bacteria (both Escherichia coli O157: H7 and Listeria innocua) and a virus (T7 bacteriophages) are both more than 99.99%. Wu, R. et al. [82] anchored oxidized dextran on the surface of melt-blown fabric and grafted the poly hexamethylene guanidine antimicrobial agent to the aldehyde group surface, thereby preparing a guanidine-loaded PP antimicrobial melt-blown fabric. The proportion of dead Escherichia coli on the surface of the melt-blown fabric increased from 17.8% to 92.0%, while the water flow rate remained as high as 5255 L/(h·m^2^). Kubacka, A. et al. [83] incorporated ${\mathrm{TiO}}_{2}$ into an isotactic PP polymer matrix to prepare a nanocomposite film by bicomponent melt-blown process, and the antibacterial activity of the oxide component was enhanced through surface charge carrier treatment. They found that when the $2\;\mathrm{wt}.\%\;{\mathrm{TiO}}_{2}$ nanocomposite film exhibited the most effective bactericidal properties against both gram-positive and gram-negative bacteria, almost all cells of two microorganisms have been rendered inactive, accounting for a log-reduction of near ca. 8 units.

#### 3.3.6. Sound Absorbing Material

The reduction of noise can be achieved through three principal methods: at the source, at the ear, and during the transmission process. Three fundamental mechanisms exist for the reduction of noise during transmission: absorption, damping, and isolation [84,85,86]. “Sound insulation” refers to reducing the penetration of noise into other areas, while “sound absorption” refers to reducing the reflection and energy of sound on the surfaces. The sound absorption coefficient of a material depends on the degree of presence of air molecules on the surface of the material and in the pores [87,88,89]. As illustrated in Figure 7, when sound waves are incident vertically on the surface of the material, a part of them are reflected directly from the material surface, while the remainder are transmitted into the interior of the material through capillaries that permeate the outside. Most of the energy is absorbed due to scattering and vibration of the fibers, and heat exchange between the air and the material also causes attenuation of the sound wave energy. The sound absorption of porous non-woven fabrics is the consequence of the combined effects of air viscosity and heat transfer.

Bhat, G. and Messiry, M.E. [90] employed cotton, polyester boards and PP fibers to create nonwoven materials by traditional melt-blown process and tested their sound absorption properties in the frequency range of 100–1500 Hz. The results demonstrated that PP microfiber melt-blown nonwoven fabric samples exhibited good sound absorption properties throughout the entire frequency range, the noise reduction coefficient of the layer using melt-blown fine fibers can reach 0.8. And the use of multi-layer samples could also enhance the sound absorption performance.

Studies have shown that the sound absorption coefficient of pure PP can be modified by the appropriate selection of the filler introduced into the PP matrix. The sound absorption coefficient of the composite fiber prepared from jute fiber and PP by bicomponent melt-blown process can reach 0.54 in the range of 100–2500 Hz [91]. Shen, J. et al. [92] used bicomponent melt-blown process to prepare JB/PP composite material, which they used to fabricate micro-perforated composite materials with various perforation characteristics. These economical and environmentally friendly materials all exhibit sound absorption coefficients exceeding 0.8 in the range of 500–1500 Hz.

Çelikel, D.C. and Babaarslan, O. [93] showed that increasing the base weight would improve the sound absorption performance of multi-layer nonwovens, and the sound absorption performance of three-layer nonwovens with bicomponent fiber as the outer layer was better than that of homogeneous nonwovens. Çinçik, E. and Aslan, E. [94] developed a three-layer composite material by using recycled polyester-based thermobonded nonwovens as the outer layer and melt-blown nonwovens from PP and polybutylene terephthalate as the inner layer. The results indicate that the sound absorption performance of nonwoven composite materials containing melt-blown layers is comparable to, or even better than, that of composites containing nanofibers and existing materials. The composite material’s maximum sound absorption coefficient of 0.46 was obtained for 630 Hz, 0.71 for 800 Hz, and 0.74 Hz for 1000 Hz sound frequencies, respectively. Moreover, 0.77–0.98 sound absorption values were also acquired in the range of 1250–3150 Hz, whereas 0.99–1 in the range of 4000–6300 Hz with the developed nonwoven composite structures.

#### 3.3.7. Battery Separator Material

With the rapid expansion of the new energy market, lithium (Li) ion batteries have gained widespread application due to their high energy density and fast charging/discharging capabilities. The battery separator, which keeps the anode and cathode electrodes separate, plays a crucial role in ensuring the free movement of Li ions while preventing short circuits [95,96,97]. PP-based melt-blown nonwoven materials have high electrolyte absorption rate and small shrinkage rate, which can be used as battery separators. And thermal stability, mechanical strength, and ionic conductivity are critical parameters for assessing the safety and operational performance of battery separators. These factors also represent the primary research focuses regarding battery separator melt-blown materials [98,99,100].

In the process of preparing the composite separator, Zhang, C. et al. [101] covered single side of the formed PP melt-blown nonwoven material surface with nano ${\mathrm{SiO}}_{2}$ particles, which enhanced the ionic conductivity, thermal stability of the material, and improved the safety of the battery. When the coating solution reaches 15wt.%, the C-rate performance of separator reaches a higher value of 152mAh/g at 0.2 C and 89 mAh/g at 5 C. On the other hand, it has been noted in the literature that the structural changes of polyolefin separators throughout their service life can have a significant impact on the operational efficiency and safety performance of Li ion batteries. But the relationship between them remains unknown [102].

PP-based melt-blown materials have also been utilized in Li-S batteries; but the problem of continuous migration of soluble polysulfides through the separator must be addressed first [103]. Zhu, W. et al. [104] used multiwall carbon nanotubes loaded with ${\mathrm{CeO}}_{2}$ nanoparticles to cooperatively modify commercial PP separators. Relying on the high polysulfide affinity of ${\mathrm{CeO}}_{2}$ and the physical adsorption capabilities of the loaded nanotubes, the shuttle effect of polysulfides was effectively mitigated. Thereby, the Li-S battery showed stable discharge performance with a capacity of 520.7mAh/g retained after 300 cycles.

## 4. Conclusions and Outlooks

This paper provides a comprehensive overview of recent research advancements in melt-blown nonwovens through literature analysis, briefly introducing the melt-blowing process and the manufacturing principles of micro/nano-fibers. Furthermore, we focus on the current applications of PP melt-blown nonwovens in areas such as media filtration and oil–water separation.

PP melt-blown materials have garnered significant attention in filtration and oil–water separation due to their exceptional adsorption capacity, lightweight nature, excellent processability, and cost-effectiveness. In recent years, with in-depth studies on melt-blowing feedstocks and manufacturing techniques, the functional applications of polypropylene melt-blown materials have been further expanded. The application of surface modification and composite reinforcement technologies has enhanced their separation efficiency for fine particles, organic pollutants, and metal ions.

However, polypropylene melt-blown materials still face challenges in large-scale industrial applications. For instance, their inherent hydrophobicity limits interactions with certain pollutants, necessitating further modifications to improve selectivity and adsorption efficiency. Additionally, the long-term environmental impact and recyclability issues of PP-based materials have raised concerns about sustainable alternatives and optimized disposal strategies. The development of biodegradable PP composites and innovative polymer processing technologies may offer viable solutions.

Future research will continue to prioritize the scalable application of functionalized. polypropylene melt-blown materials. Meanwhile, advancements in advanced nanomaterials and bio-based additives may further broaden their application forms. Furthermore, optimizing process parameters, adopting environmentally friendly manufacturing processes, and promoting circular economy principles can also help mitigate environmental impacts.

## Figures and Tables

**Figure 1 polymers-17-01013-f001:**
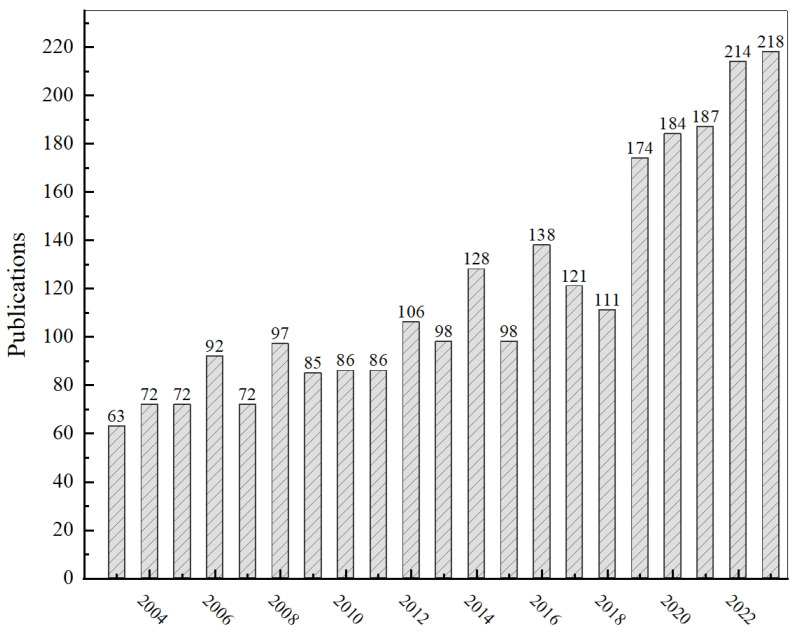
Statistical chart of the number of melt-blown nonwoven material publications in the last years. (data from WOS core database).

**Figure 2 polymers-17-01013-f002:**
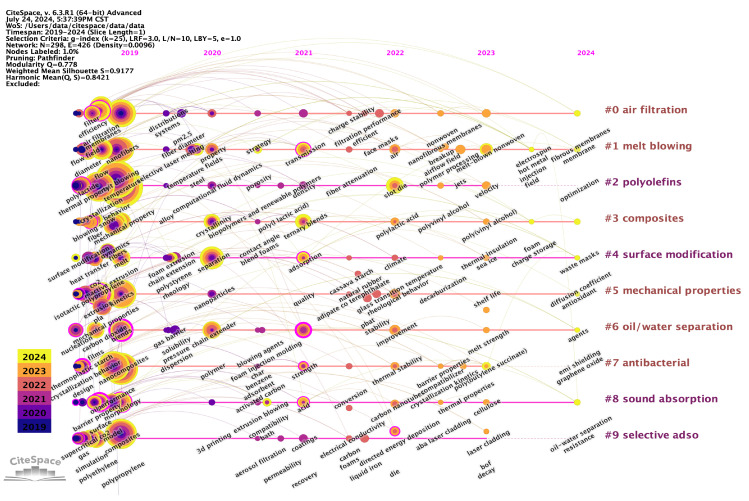
Keyword time mapping of research results on melt-blown nonwoven materials over the past decade.

**Figure 3 polymers-17-01013-f003:**
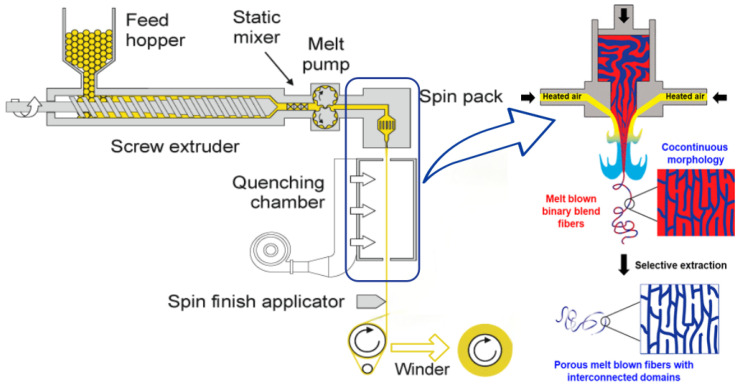
Schematic diagram of melt-blown process flow and principle [3].

**Figure 4 polymers-17-01013-f004:**
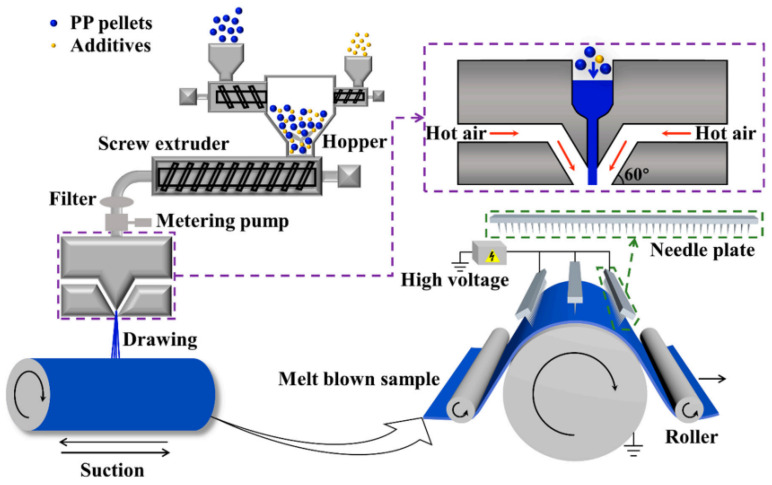
Schematic diagram of bielectret corona charging process [40].

**Figure 5 polymers-17-01013-f005:**
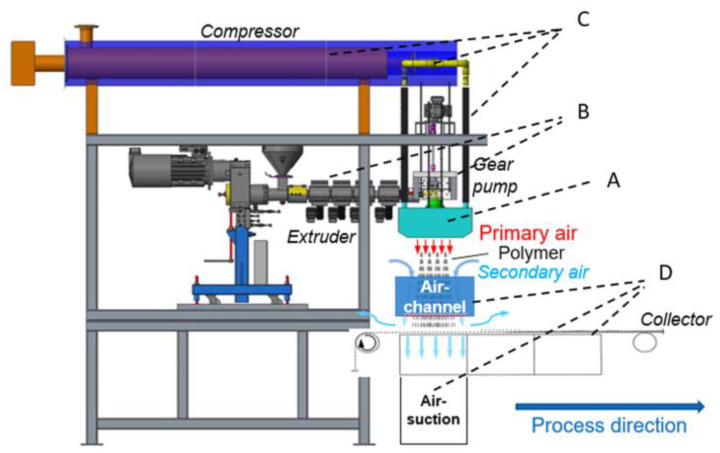
Nanoval split spinning nonwoven material process flow diagram: the Nanoval spinning beam itself (A), an extrusion system (B), a system for the supply of the hot process air (C), and the deposition and air treatment system (D) [45].

**Figure 6 polymers-17-01013-f006:**
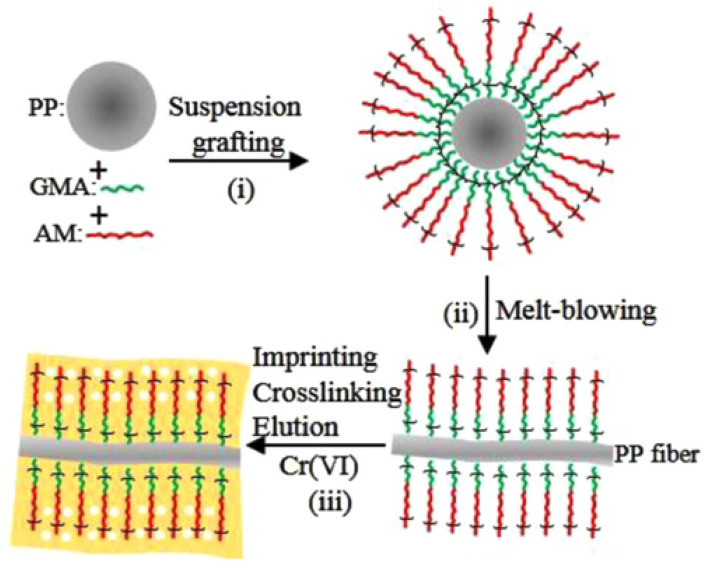
Schematic diagram of the synthesis process of Cr(VI)-imprinted fibers [67].

**Figure 7 polymers-17-01013-f007:**
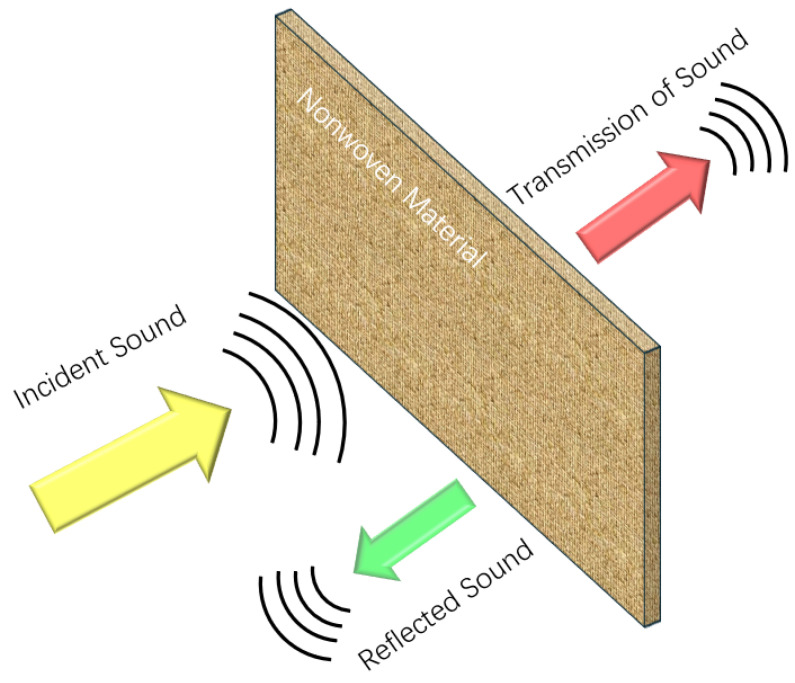
Mechanism of “sound absorption” of the nonwoven material.

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
