# Peer review of "Advancements in Research and Applications of PP-Based Materials Utilizing Melt-Blown Nonwoven Technology"

_polymers, 2025, doi:10.3390/polym17081013_

Round 1
Reviewer 1 Report
Comments and Suggestions for Authors
The paper introduces the melt-blown process and the principle of micro/nano-fiber manufacturing, reviews the development of melt-blown raw materials and melt-blown process technology, and summarizes the status of melt-blown nonwoven materials in various fields. The paper also considers the potential applications of melt-blown nonwoven materials in several other areas, including media filtration, oil- water separation, heavy metal ion adsorption, organic pollutant removal, healthcare, and new energy. It analyses the current problems and development directions in these areas, to provide a certain reference for the research and application of melt-blown nonwoven materials.
After reviewing this manuscript, I found it an interesting work. However, there are some remarks have to be considered before recommending it for publication.
- The melt-blown process is suggested to be described in more detailed manner and introduce the characteristic parameters affect it.
- In micro/nano-fiber manufacturing section, the more recent advancements, which is introduced with higher efficiency and less costs, shall be presented and compared.
- The main differences and
- The fundamental advantages and disadvantages of both melt-blown raw materials and melt-blown process technology shall be described in detail comparing with traditional materials and processes.
- There are also various applications of the melt-blown nonwoven materials in chemical industries, which shall be presented with new advanced technologies.
- Some complex sentences arise through the manuscript. Shorter and more comprehensive ones will be better.
- The conclusion section has to be rewritten.
- Recent and relevant references published in 2024 and 2025 are recommended to be cited here.
The paper introduces the melt-blown process and the principle of micro/nano-fiber manufacturing, reviews the development of melt-blown raw materials and melt-blown process technology, and summarizes the status of melt-blown nonwoven materials in various fields. The paper also considers the potential applications of melt-blown nonwoven materials in several other areas, including media filtration, oil- water separation, heavy metal ion adsorption, organic pollutant removal, healthcare, and new energy. It analyses the current problems and development directions in these areas, to provide a certain reference for the research and application of melt-blown nonwoven materials.
After reviewing this manuscript, I found it an interesting work. However, there are some remarks have to be considered before recommending it for publication.
- The melt-blown process is suggested to be described in more detailed manner and introduce the characteristic parameters affect it.
- In micro/nano-fiber manufacturing section, the more recent advancements, which is introduced with higher efficiency and less costs, shall be presented and compared.
- The main differences and
- The fundamental advantages and disadvantages of both melt-blown raw materials and melt-blown process technology shall be described in detail comparing with traditional materials and processes.
- There are also various applications of the melt-blown nonwoven materials in chemical industries, which shall be presented with new advanced technologies.
- Some complex sentences arise through the manuscript. Shorter and more comprehensive ones will be better.
- The conclusion section has to be rewritten.
- Recent and relevant references published in 2024 and 2025 are recommended to be cited here.
Reviewer 2 Report
Comments and Suggestions for Authors
The article provides an overview of polypropylene nonwoven materials by melt blowing technology, and includes: methods for producing nonwoven mats, modifications of PP materials, and the application of polypropylene nonwovens. The review article is short both in content and in the list of references. Usually, short reviews briefly provide information on the current trend and achievements in science over the past 2-3 years. The manuscript submitted for consideration does not contain this. The text periodically contains duplicate sentences that should be said once in the introduction. To summarize, the manuscript needs to be significantly revised both in terms of meaning and in list of references.
Major Remarks:
1. All abbreviations PP, PPS, PLA, etc. in section titles should be given without abbreviations. Also, abbreviations occurring in the text for the first time or once should be expanded
2. Page 3 line 80-81. Of the entire range of listed polymers, biodegradation is characteristic only of polylactide. In addition, reference [7] does not characterize these properties.
3. The title of the work highlights PP-materials, but the authors described other materials in section 3.1 as materials for nonwovens. It is necessary to remove these sections or provide the results of using composites based on polypropylene with the addition of other polymers. In particular, Sections 3.1.2 and 3.1.4 do not contain data on PP-materials.
4. Section 3.2 is devoted to manufacturing technologies. Here it is necessary to clearly describe how each of the methods technically differs from each other. Sometimes, the Authors use general phrases that have no scientific or technical meaning.
5. In section 3.2.3, what does the word "composite" in the title mean? There is no mention in the text.
6. Section 3.3. Each section begins with words about the small -sized pored structure and fine fibers typical of nonwoven materials. The description of the material should be given once in the introduction when describing the relevance of the work. Expansion of the text by repetitions is inappropriate.
7. Page 11 line 359-376. The text and results do not describe the PP-materials.
8. Qualitative descriptions of results are occasionally encountered, although clear quantitative values are required. For example, Section 3.3.6. line 449: "Good sound absorption", line 513: "noticeable increase in absorption coefficient"... and others. Results obtained from the reference should be quantitative.
9. I propose to combine section 4 and the conclusion.
10. The list contains 94 references, of which only two articles are from 2024, and ten articles are from the year 2023. Authors need to add more recent references from 2025 and 2024.
Minor remarks:
1. Figure 1, start numbering the years from the smallest to the largest.
2. Many semantic repetitions. For example, page 9, line 296, the range of meanings is given numerically and verbally as a wide range of diameters.
3. The title of Section 3.3. It is better to remove the word "main"
Round 2
Reviewer 2 Report
Comments and Suggestions for Authors
The list of references should be updated. References refer to the old version of the manuscript.
Author Response
Comment: The list of references should be updated. References refer to the old version of the manuscript.
Response: Thank you for pointing this out. We found that the references in the PDF version of the manuscript differ from those in the LaTeX source files. We have updated the references in the PDF version accordingly, and the references in the LaTeX compressed file are the correct version.